# Does the Development of Information and Communication Technology and Transportation Infrastructure Affect China's Educational Inequality?

**Peng Zhou [1], Fengwen Chen [2,\*], Wei Wang [3,\*], Peixin Song [2] and Chenliang Zhu [4]**

1   National Engineering Research Center for E-Learning, Central China Normal University, Wuhan 430079, China; stevezp@mail.ccnu.edu.cn
2   School of Economics and Business Administration, Chongqing University, Chongqing 400030, China; songpeixin@cqu.edu.cn
3   School of Management and Engineering, Nanjing University, Nanjing 210093, China
4   School of Information Management, Central China Normal University, Wuhan 430079, China; zhuchenliang@mails.ccnu.edu.cn
\*   Correspondence: chenfengwen@cqu.edu.cn (F.C.); wwqd2hs@smail.nju.edu.cn (W.W.); Tel.: +86-0236-5106-482 (F.C.)

**Abstract:** Educational inequality is an important factor in the development of human capital, and limits the output of regional economic activities. The unequal distribution of educational resources has become a hot topic noticed by the public, and has restricted sustainable economic growth. This paper provides a better understanding of educational inequality, and explores the impacts of information and communication technology (ICT) and transportation infrastructure on the distribution of educational resources. The panel data models are constructed to discuss the relationship among ICT, transportation infrastructure, and educational inequality, using the data of 31 provinces in China from 2006 to 2016. The empirical results show that there is a positive relationship between ICT and educational inequality, while transportation infrastructure can restrain the unequal distribution of educational resources. Moreover, there is a significant inverted U-shaped relationship between transportation infrastructure and educational inequality. Since China's education reform in 2010, the relationship among ICT, transportation infrastructure, and educational inequality has been significantly changed, as well as the influence mechanism of ICT. In addition, transportation infrastructure in China western regions can effectively alleviate the problem of educational inequality, and its impact will increase with the growth of transportation investments. It is necessary to consider the rational allocation of educational resources, and this is essential to relieve the problem of educational inequality. Therefore, our results demonstrate the key roles of information technology and transportation network in the field of education, and provide some new ideas for the solution of educational inequality.

**Keywords:** educational inequality; ICT; transportation infrastructure; sustainable economic growth

## 1. Introduction

With the continuous development of the global economy, educational outcomes have become an important factor in promoting regional economic growth. Educational attainment as the representation of educational outcomes measures the degree of education a person has completed, and its development relies on the effective utilization of educational investment. During the educational process, the economic value of labor force can be reflected in the human capital, which is the stock of educational investment. Considering the role of the labor force in regional economic activities, there is a close relationship between educational factors and the labor market, while the educational attainment of

the labor force can directly determine the efficiency of economic output, which is the result of economic activities through utilization of external resources. For many developing countries, the development of regional economy still depends on labor-intensive industries, and human capital is essential to support national strength. From the perspective of human capital, the literacy and capability of labor force are mainly promoted by better educational quality, and the unequal distribution of educational resources may lead to regional inequality of labor force literacy and ability. The inequality of educational resources can be directly reflected in educational inequality, and then educational groups cannot obtain equal educational opportunities [1]. The emergence of educational inequality has not only become a problem in the field of education, but has also hindered the development of real economy. In addition, educational inequality can be affected by the gap between the rich and the poor. The accessibility and affordability of educational resources have struggled to meet social requirements, so fairness of education cannot be guaranteed. Compared to other developing countries, China has provided a large amount of labor to the global market, and plays an important role in labor-intensive industries. However, the Chinese government has been trying to upgrade industrial structure from labor-intensive to technology-intensive, and this transformation depends on labor capability. In China's different regions, the unbalanced distribution of resources leads to the emergence of educational inequality, and hinders industrial transformation [2]. Facing these difficulties, educational inequality has become an important topic of concern to government regulators and researchers, and its solution will be a key factor in promoting sustainable economic growth.

In contrast to the traditional education model, access to educational resources has gradually got rid of spatial limitations. Online education can provide more educational opportunities for the public, but this relies on the development of information and communication technologies (ICT). ICT provides users with more convenient information services, and links social groups. ICT helps users get more information resources, while they need to pay for the application of this technology. Considering the affordability of ICT, not all learners can take advantages of this service, which enlarges the inequality between users in resource acquisition. Regional education inequality may also be enlarged by ICT. Furthermore, the distribution of educational resources may be also influenced by space factors. In order to get rid of the constraints of geographical factors, transportation infrastructure effectively shortens the distance between different regions, and provides users with convenient transportation services. The development of a transportation network lengthens the distance of labor migration, and shortens the distance between urban and rural areas. Given the characteristics of China's population policy, the hukou system hinders the development of human capital, and restricts the educational opportunities of the labor force [2]. Faced with the limitations of the hukou system, the development of the transportation network can change the distribution of human resources, and allow more labor to obtain educational opportunities, thereby improving the availability of educational resources. Combining the features of educational resources and educational opportunities, exploring the relationship among ICT, transportation infrastructure, and educational inequality has become a key element for the rational allocation of educational resources, and this can also promote industrial transformation, guaranteeing sustainable development of the regional economy.

This paper discusses the roles of ICT and transportation infrastructure in the educational field, and its contributions are mainly reflected in the following points. First of all, the role of ICT in educational inequality is neglected, and improving the availability and affordability of information services will directly change the distribution of educational resources. Secondly, transportation infrastructure can improve the constraints of geographical factors and population policy, and help more social groups obtain fair educational opportunities. Finally, this paper takes into account the urban–rural gap in developing countries, and explores educational inequality from the perspectives of information technology development and transportation infrastructure investment. The findings can help government regulators design reasonable education policies, maintaining equal distribution of educational resources and the sustainability of regional economic development.

The structure of this paper is as follows: Section 2 will introduce the literature review and research hypotheses. Section 3 will introduce variables and models, including data sources, variable construction, and empirical models. Section 4 will introduce research results, including descriptive statistics and empirical results. Section 5 will present a discussion. Section 6 will present our conclusions and recommendations.

## 2. Literature Review and Research Hypothesis

### 2.1. Educational Inequality

Educational inequality is the starting point of many studies, and provides some explanation for a variety of economic issues. Based on the characteristics of educational inequality, some existing research has explored the micro-features and macro-impacts of this problem, and discussed the relationship between educational inequality and economic topics [1,3,4]. In terms of the micro-features of educational inequality, the structural characteristics and family characteristics of educational groups become important factors in promoting the unequal distribution of educational resources, and these studies have been concentrated on the individual perspective [3,5]. In terms of the macro-impacts of educational inequality, there is a close relationship between educational resources and economic activities, and this relationship plays an important role in the formulation of economic policies and educational policies [1,4].

In order to explore the influence factors of educational inequality, many researchers have made analyses of the impact of micro-features on educational inequality from the individual perspective. Picard and Wolff (2010) used a variety of methods to measure educational inequality, finding that the distribution of educational resources and the characteristics of family members could promote the emergence of educational inequality [3]. McKenzie and Rapoport (2011) explored the influencing factors of educational inequality from the perspective of migration, pointing out that migration would reduce the average educational attainment, while gender and age factors could promote the issue of educational inequality [6]. Moreover, the age and income of family members could affect the outcomes of public education, while tax policies could reduce the impact of educational inequality on social stability [7]. Valerio Mendoza (2018) discussed the problem of educational inequality in China, pointing out that the inequality in secondary education and higher education could aggravate the graveness of educational inequality, and that family factors could also promote this problem [5]. Family and social factors could contribute to the emergence of educational inequality, but individual income became a determining factor in the inequality of educational opportunities and educational outcomes [8,9]. Tselios (2014) explored the impact of income inequality on educational inequality in Europe, pointing out that the differences in the levels of individual income could motivate educational inequality [10]. Furthermore, there was a significant relationship between educational attainment and income level, and then educational inequality would be impacted by the gap between the rich and the poor [11]. The efficiency of fiscal expenditure could mediate the relationship between income level and educational inequality, while individual income became the main factor in obtaining the educational opportunities [12].

In comparison with studies of micro-features, the relationship among educational inequality, economic factors, and social factors became a hot topic noticed by researchers in different fields. Rodríguez-Pose and Tselios (2010) explored the relationship between educational inequality and economic growth, pointing out that the improvement of average educational attainment could not relieve the potential problems caused by educational inequality, and educational inequality would hinder regional economic development [13]. Hanushek and Woessmann (2012) analyzed the relationship between educational inequality and economic development, finding that educational inequality could affect the skills of the labor force, and that this relationship was more pronounced in developing countries [14]. Ferreira and Gignoux (2014) measured educational inequality from the perspective of educational outcomes and educational opportunities, suggesting that educational

inequality in Europe and Latin America was relatively high, and that the unbalanced distribution of educational resources would inhibit the development of regional economy [1]. The effect of educational inequality could be controlled by the development of regional economy, while education policy could alleviate educational inequality in economically backward areas [15]. In addition, education reform played different roles in various educational groups, and it was difficult to achieve an equilibrium state between educational input and educational output [4]. Meschi and Scervini (2014) discussed the relationship between schooling expansion and educational inequality, suggesting that inequality in higher education was more serious, and the length of compulsory education could relieve the problem of educational inequality [16]. Moreover, higher educational attainment could lower mortality, and educational expansion would provide more educational resources for the whole society while reducing the negative impact of educational inequality [17].

According to the existing research on educational inequality, the influencing factors of this problem are concentrated on micro-features, while macro factors have lacked attention from researchers, which has limited the generality and practicality of the solutions offered to educational inequality. In order to better allocate educational resources, it is necessary to explore educational inequality from the macro perspective.

## 2.2. ICT and Educational Inequality

ICT is an important concept in the field of information science, and represents the development of information technology. The development of ICT can help more users obtain Internet information and improve the convenience and availability of information services. Moreover, information technology plays an important role in the field of education, but the relationship between ICT and educational inequality is still at the exploratory stage.

Considering the role of ICT in the educational field, researchers have often conducted analyses of information technology from the perspective of application. Angus et al. (2004) discussed the advantages and disadvantages of ICT, finding that the availability of Internet information could affect the distribution of educational resources and relieve the seriousness of educational inequality [18]. Nachmias et al. (2010) analyzed the application of ICT in the educational field, pointing out that ICT could influence the practical ability of educational groups [19]. In addition, the development of ICT promoted cultural exchange, and strengthened the relationship between cultural capital and educational resources [20]. Tondeur et al. (2011) explored the relationship between socioeconomic status and ICT from the perspective of cultural capital, pointing out that users with more social resources would pay more attention to ICT, and get more educational resources [21]. Nath and Liu (2017) analyzed the role of ICT in imports and exports, suggesting that ICT could promote technology imports, and improve the unbalanced distribution of labor capital in different regions [22]. However, ICT might reduce the utilization efficiency of educational resources [23]. Ayanso and Lertwachara (2015) analyzed the relationship between the development and affordability of ICT, finding that the pricing process of ICT limited users' experience and decreased the efficiency of information acquisition [24]. In terms of ICT affordability, there was a close relationship between information technology and the urban–rural gap, and digital resources could improve the efficiency of information acquisition, while users in backward areas would have difficulty enjoying the benefits of ICT [25]. Park (2017) explored the application of ICT from the perspective of the urban–rural gap, finding that ICT in rural areas was difficult to use to its advantage, and that it increased the gap between urban and rural areas in educational attainment [26]. Digital inequality in ICT could hinder the flow of information resources, and this inequality would promote the unbalanced distribution of urban and rural resources [27]. During the exploring process of the relationship between ICT and educational inequality, Billon et al. (2018) pointed out that educational inequality could mediate the relationship between Internet use and economic development, but this inequality in developing countries would restrict the effect of ICT [28].

Considering the rapid development of information technology, less research has directly explored the relationship between ICT and educational inequality. It is worth noting that ICT as a double-edged

sword can widen the gap between urban and rural areas, and the balanced distribution of educational resources is difficult to achieve. Therefore, we propose the following hypotheses:

**Hypothesis 1a:** *There is a positive correlation between ICT and educational inequality;*

**Hypothesis 1b:** *There is a nonlinear relationship between ICT and educational inequality.*

### 2.3. Transportation Infrastructure and Educational Inequality

Transportation infrastructure is an important factor in promoting regional economic development, and the development of a transportation network shortens the distance between different regions and reduces the geographical restrictions on labor migration. Considering the role of transportation infrastructure in economics, there is still scant research on the effect of transportation infrastructure in the educational field. Therefore, exploring the relationship between transportation infrastructure and educational inequality could become the key element in promoting the development of human capital.

In order to obtain the relationship between transportation infrastructure and economic development, researchers have often discussed this relationship from the perspective of human capital and employment. Nunn (2004) found that transportation infrastructure in different countries could improve labor migration and promote the development of regional economy [29]. Schaffer and Siegele (2009) made an analysis of the relationship between transportation infrastructure and human capital, pointing out that transportation infrastructure could promote the ability of the labor force, and improve the average educational attainment [30]. The transportation network could affect the accessibility of jobs, while educational attainment increased the employment rate [31]. From a long-term perspective, transportation infrastructure not only shortened the distance between regions, but also changed the distribution of urban and rural resources [32]. Hannum and Wang (2006) pointed out that some geographical factors could affect the distribution of educational resources [33]. Transportation infrastructure could provide more job opportunities for rural labor, and increase income level in rural areas while reducing the gap between urban and rural areas [34,35]. Li and DaCosta (2013) explored the impact of transportation infrastructure on income inequality, pointing out that the transportation network could relieve the gap between the rich and the poor [36]. In addition, higher educational attainment would promote transportation sustainability [37]. Teulings et al. (2018) found that transportation infrastructure could improve employee welfare, but this relationship was limited by their educational attainment [38]. Given the role of transport infrastructure in the utilization efficiency of educational resources, economic development and population policy might restrict the role of transportation network in different regions [39]. Liu (2005) analyzed the impact of educational inequality on the urban–rural gap in China, suggesting that the unbalanced distribution of educational resources could directly strengthen the gap between urban and rural areas, while the hukou system would enhance this relationship [40]. Moreover, the hukou system limited the educational attainment of family members, and created more obstacles in alleviating the problem of educational inequality [2].

Based on the role of transportation infrastructure in educational field, increasing the scope of labor migration could overcome the disadvantages of the traditional education model, and encourage social groups to enjoy fair educational opportunities, reducing the restrictions of educational inequality. Therefore, we propose the following hypotheses:

**Hypothesis 2a:** *There is a negative correlation between transportation infrastructure and educational inequality;*

**Hypothesis 2b:** *There is a nonlinear relationship between transportation infrastructure and educational inequality.*

## 3. Variables and Models

### 3.1. Data Source

In order to obtain the relationship among ICT, transportation infrastructure, and educational inequality, this paper selected the data of 31 provinces (including 4 municipalities) in China from 2006 to 2016 as the overall sample. To ensure the reliability and accuracy of data, the data of educational inequality mainly came from the National Bureau of Statistics of China and the education statistics yearbooks of China [41,42]. The education statistics yearbooks of China mainly reflect the development of educational system in China, and include the data of students and institutions at different educational degrees. For the measurement of educational inequality, the average years of schooling was computed by the data from the National Bureau of Statistics of China, and the population at different educational degrees came from chapter one (named as the development of education) of the education statistics yearbooks. At the same time, the data of ICT and transportation infrastructure mainly came from the statistical yearbooks and transportation statistical yearbooks of various provinces [41]. The transportation statistical yearbooks include the data of transportation density and transportation investment, and the growth of transportation investment is computed by the data of Construction of Transportation Infrastructure in this kind of yearbook. The data of control variables were derived from the National Bureau of Statistics of China, the financial yearbooks, and the statistical yearbooks of various provinces [41,43]. The financial yearbooks provide the data of government revenue and expenditure, and reflect the fiscal situation of different provinces.

### 3.2. Variable Construction

#### 3.2.1. Dependent Variable

Educational inequality, as the dependent variable, represents the unequal distribution of educational resources in different regions. In the measurement of educational inequality, the inequality of educational outcomes can better reveal the characteristics of educational inequality at the macro level. Referring to Zhang and Kanbur (2005) and Billon et al. (2018), we used the educational attainment Gini coefficient to measure the educational inequality of different provinces in China [28,44]. The calculation method of educational inequality is shown in Formula (1):

$$EduIneq = \frac{1}{\mu} \sum_{k=2}^{n} \sum_{j=1}^{k-1} |x_k - x_j| p_k p_j \qquad (1)$$

where *EduIneq* represents the educational attainment Gini coefficient that measures educational inequality; $\mu$ represents the average years of schooling; $p_k$ and $p_j$ represent the proportion of the population at different levels of education; $x_k$ and $x_j$ represent the number of years of education at different educational levels; and $n$ represents the number of educational levels, and five educational levels are selected in this paper, including no schooling, primary education, secondary education, high school education, and higher education. A higher value of this variable indicates that the unbalanced educational resources is more serious.

#### 3.2.2. Independent Variables

Information and communication technology (ICT), as an independent variable, represents the development of information technology and the availability of information resources. In order to better understand the development of ICT at the macro level, it needs to consider the indicators in different dimensions, thereby improving the reliability and usability of the measurement method. Referring to Nath and Liu (2017), we used the development index of information and communication technology constructed by the International Telecommunication Union (ITU) to measure the ICT of different provinces in China [22,45]. The ICT Development Index can reveal the development of information technology in different regions from multiple dimensions, including ICT access, ICT use, and ICT

skills, as shown in Table 1. Considering the values of other variables, we converted the value of ICT Development Index to the range from 0 to 1, with 0 indicating the minimum level of ICT development and 1 indicating the maximum level of ICT development.

**Table 1.** The construction of the ICT Development Index.

| Dimension | Indicators | Indicator Weights | Dimension Weights |
|---|---|---|---|
| ICT access | Fixed telephone subscriptions per 100 inhabitants | 25% | 40% |
| | Mobile cellular telephone subscriptions per 100 inhabitants | 25% | |
| | Percentage of households with a computer | 25% | |
| | Percentage of households with Internet access | 25% | |
| ICT use | Percentage of individuals using the Internet | 50% | 40% |
| | Fixed broadband subscriptions per 100 inhabitants | 50% | |
| ICT skills | Mean years of schooling | 33% | 20% |
| | Secondary gross enrolment ratio | 33% | |
| | Tertiary gross enrolment ratio | 33% | |

Transportation infrastructure, as an independent variable, represents the development of the regional transportation network and the accessibility between cities. In the measuring process of transportation infrastructure, it was necessary to consider both the development of transportation network and the sustainability of transportation investment. Based on the development of China's transportation infrastructure, Tong and Yu (2018) discussed different measurement methods of transportation infrastructures, but it is difficult to reflect how governments focus on transportation infrastructure from the perspectives of capital and density [46]. Considering the close relationship between transportation infrastructure and regional economy, the investment in transportation network plays an important role in the GDP competition among different provinces in China. Therefore, we referred to the method of Yu et al. (2012), and measured the variable of transportation infrastructure by using the proportion of growth of transportation investment to the regional GDP [47].

### 3.2.3. Controlled Variables

Macroeconomic factors have always played an important role in many studies of educational inequality, as well as social factors [28,33]. In order to be consistent with previous literature on educational inequality, we selected inflation, unemployment rate, government intervention, education expenditure, and fixed assets investment as the controlled variables to explore the influence of ICT and transportation infrastructure on educational inequality.

### 1. Inflation

Inflation represents the consumption ability of social groups in different regions. The degree of inflation could influence the entities of economic activities, and play an important role in maintaining social stability. The consumer price index was used to measure the degree of inflation and control for the impact of social factors on educational inequality.

### 2. Unemployment rate

The unemployment rate can intuitively reflect the actual demand of the labor market, and become the key factor in individual income. Changes of unemployment rate will affect social stability and limit the output of economic activities. The proportion of unemployed individuals was used to

measure the regional unemployment rate, in order to control for the impact of social factors on educational inequality.

3. Government intervention

Government intervention represents the role of local government in the development of regional economy. Given the economic competition among different provinces, government intervention could determine the structure of regional industries and the goal of economic development. From the differences of regional industrial structure, the economic development strategies of local governments will affect the output of economic activities, as well as the distribution of educational resources. The proportion of government fiscal expenditure to the regional GDP was used to measure government intervention, to control for the impact of economic factors on educational inequality.

4. Education expenditure

Education expenditure is an important component of government expenditure, and represents government investment in the educational field. Although the development of regional economy can promote education expenditure, the impact of the demographic factor should not be ignored. The proportion of fiscal expenditure on education to the regional population was used to measure education expenditure, and to control for the impact of economic factors on education inequality.

5. Fixed assets investment

Fixed asset investment represents the government investment in infrastructure construction. From the perspective of regional economy, infrastructure construction can provide abundant guarantees for economic activities, and can also improve the quality of life. Compared to other types of resources, the application of educational resources is more dependent on the development of infrastructure. The proportion of total fixed assets investment to the regional GDP was used to measure fixed asset investment, in order to control for the impact of economic factors on educational inequality.

The introductions and definitions of all variables are shown in Table 2.

**Table 2.** Variable definitions and measures.

| Abbreviations | Variables | Definitions |
| --- | --- | --- |
| EduIneq | Educational inequality | Educational inequality is the unbalancing distribution of educational resources, and is measured by the Gini coefficient on education. |
| ICT | Information and communication technology | Information and communication technology is measured by the ICT development index. |
| TRANS | Transportation infrastructure | Transportation infrastructure is measured by the proportion of growth of transportation investment to the regional GDP. |
| INFL | Inflation | Inflation is measured by consumer price index. |
| UNEM | Unemployment rate | Unemployment rate is measured by the proportion of the unemployed individuals to the all individuals in the labor force. |
| GOV | Government intervention | Government intervention is measured by the proportion of government fiscal expenditure to the regional GDP. |
| EDU | Education expenditure | Education expenditure is measured by the proportion of fiscal expenditure on education to the regional population. |
| FIV | Fixed assets investment | Fixed assets investment is measured by the proportion of total fixed assets investment to the regional GDP. |

*3.3. Empirical Models*

In the exploring process of educational inequality, ICT and transportation infrastructure will have a corresponding impact on the distribution of educational resources from the information dimension and the space dimension. In order to obtain the relationship among ICT, transportation infrastructure and educational inequality, the panel data model was used to analyze this relationship. Because the unbalanced distribution of educational resources is difficult to alleviate in a short time, we used the dynamic panel data model to empirically test the research hypotheses and control the province fixed

effects and the year fixed effects. The dynamic panel data model can better explain the dynamic features of educational inequality, and the lag item of educational inequality will show the continuity of unequal distribution of educational resources. Furthermore, controlling the province fixed effects and the year fixed effects can properly demonstrate the impact of ICT and transportation infrastructure from the macro perspective.

$$
\begin{aligned}
EduIneq_{i,t} = {} & c + \alpha_0 EduIneq_{i,t-1} + \alpha_1 ICT_{i,t} + \alpha_2 TRANS_{i,t} + \beta_1 INFL_{i,t} + \beta_2 UNEM_{i,t} + \beta_3 GOV_{i,t} + \\
& \beta_4 GOV_{i,t-1} + \beta_5 EDU_{i,t} + \beta_6 EDU_{i,t-1} + \beta_7 FIV_{i,t} + \beta_8 FIV_{i,t-1} + \mu_i + \omega_t + \varepsilon_{i,t}
\end{aligned} \tag{2}
$$

Formula (2) is the basic model of this paper. Considering the continuity of economic activities, the lagged items of partial control variables are introduced to better control the impact of economic factors on educational inequality. $EduIneq_{i,t}$ indicates the educational inequality of $i$ province in $t$ year; $ICT_{i,t}$ indicates the ICT of $i$ province in $t$ year; $TRANS_{i,t}$ indicates the transportation infrastructure of $i$ province in $t$ year; $INFL_{i,t}$ indicates the inflation of $i$ province in t year; $UNEM_{i,t}$ indicates the unemployment rate of $i$ province in $t$ year; $GOV_{i,t}$ indicates the government intervention of $i$ province in $t$ year; $EDU_{i,t}$ indicates the education expenditure of $i$ province in $t$ year; $FIV_{i,t}$ indicates the fixed assets investment of $i$ province in $t$ year; $\mu_i$ and $\omega_t$ are the province fixed effect and the year fixed effect, respectively; $\varepsilon_{i,t}$ is the error term.

Considering the research hypotheses in Sections 2.2 and 2.3, there may be a nonlinear relationship between ICT or transportation infrastructure and educational inequality. Therefore, Formula (2) is converted to Formula (3) and Formula (4).

$$
\begin{aligned}
EduIneq_{i,t} = {} & c + \alpha_0 EduIneq_{i,t-1} + \alpha_1 ICT_{i,t} + \alpha_2 ICT^2_{i,t} + \beta_1 INFL_{i,t} + \beta_2 UNEM_{i,t} + \beta_3 GOV_{i,t} + \\
& \beta_4 GOV_{i,t-1} + \beta_5 EDU_{i,t} + \beta_6 EDU_{i,t-1} + \beta_7 FIV_{i,t} + \beta_8 FIV_{i,t-1} + \mu_i + \omega_t + \varepsilon_{i,t}
\end{aligned} \tag{3}
$$

Formula (3) introduces the square of ICT and explores the nonlinear relationship between ICT and educational inequality, which could fully demonstrate the impact of information technology on the distribution of educational resources.

$$
\begin{aligned}
EduIneq_{i,t} = {} & c + \alpha_0 EduIneq_{i,t-1} + \alpha_1 TRANS_{i,t} + \alpha_2 TRANS^2_{i,t} + \beta_1 INFL_{i,t} + \beta_2 UNEM_{i,t} + \beta_3 GOV_{i,t} + \\
& \beta_4 GOV_{i,t-1} + \beta_5 EDU_{i,t} + \beta_6 EDU_{i,t-1} + \beta_7 FIV_{i,t} + \beta_8 FIV_{i,t-1} + \mu_i + \omega_t + \varepsilon_{i,t}
\end{aligned} \tag{4}
$$

Formula (4) introduces the square of transportation infrastructure and explores the nonlinear relationship between transport infrastructure and educational inequality, which could comprehensively reveal the impact of the transportation network on the distribution of educational resources.

## 4. Research Results

### 4.1. Descriptive Statistics

In our empirical analysis, we used the data of 31 provinces (including 4 municipalities) in China to explore the impact of ICT and transportation infrastructure on educational inequality. It was necessary to make a descriptive statistical analysis to exhibit the characteristics of all variables. The results of descriptive statistics are shown in Table 3.

**Table 3.** The descriptive statistics of variables.

| Variables | Observations | Mean | Median | Maximum | Minimum | Standard Deviation |
|---|---|---|---|---|---|---|
| EduIneq | 341 | 0.2287 | 0.2157 | 0.5301 | 0.1699 | 0.0535 |
| ICT | 341 | 0.2875 | 0.2406 | 0.7703 | 0.0753 | 0.1557 |
| TRANS | 341 | 0.034 | 0.0264 | 0.1905 | 0.0043 | 0.0251 |
| INFL | 341 | 0.0292 | 0.025 | 0.101 | −0.023 | 0.02 |
| UNEM | 341 | 0.0351 | 0.036 | 0.051 | 0.012 | 0.0066 |
| GOV | 341 | 0.2483 | 0.2068 | 1.3792 | 0.0837 | 0.1902 |
| EDU | 341 | 0.1341 | 0.1259 | 0.5163 | 0.0223 | 0.0822 |
| FIV | 341 | 0.702 | 0.6912 | 1.3862 | 0.2398 | 0.23 |

Among different variables, the standard deviation of the educational inequality (0.0535) was low, indicating that there are a few differences of educational inequality among different regions. In addition, the minimum of educational inequality (0.1699) was close to its mean (0.2287), while the maximum (0.5301) was twice as high as its mean (0.2287), indicating that the unbalanced distribution of educational resources in some regions has been very serious. The range between the minimum (0.0753) and the maximum (0.7557) of ICT was quite large, indicating that there are obvious differences in the availability of information resources between different regions, which would promote the unequal distribution of educational resources. However, the standard deviation of transportation infrastructure (0.0251) was low, but the range between the minimum value (0.0043) and the maximum value (0.1905) was large, which indicates that the transportation investment is concentrated in the part of provinces. In the descriptive statistical results of control variables, it can be seen that the standard deviations of government intervention and fixed assets investment were higher than other variables, which indicates that the governments of different regions have greater differences in management efficiency and development goals. In addition, the standard deviations of inflation and unemployment rate were lower than other variables, reflecting the efforts of local governments in social stability. The range between the maximum (0.5163) and minimum (0.0223) of education expenditure was quite large, and the mean (0.1341) was much higher than the minimum (0.0223), which indicates that the financial resource allocation for education in each province is quite unbalanced, and educational investment would also be affected by regional economy.

Before empirical analysis, it was necessary to take the unit root test to analyze the stationarity of all variables mentioned in Section 3.2. Four kinds of unit root test methods were selected in this paper, including LLC test, IPS test, ADF-Fisher test, and PP-Fisher test. Combined with the results of the unit root test in Table A1 (Appendix A), all variables were integrated to the first-order, and were stationary under the same conditions, ensuring the reliability and accuracy of empirical results.

*4.2. Empirical Results*

4.2.1. Regression Analysis of the Overall Sample

In order to solve the unbalanced distribution of educational resources, we selected the data of 31 provinces in China to conduct an empirical analysis of educational inequality from the perspective of developing countries. Considering the regional development of China, the overall sample was used to analyze the impact of information technology and transportation network, and the empirical models were constructed by Formula (2), Formula (3), and Formula (4) to explore the linear and nonlinear relationships among ICT, transportation infrastructure, and educational inequality. Based on the results of discriminating the type of panel data model (Table A2 in Appendix A), the fixed effect model should be established to better explore the impact of ICT and transportation infrastructure on educational inequality. The regression results of overall sample are shown in Table 4.

**Table 4.** Regression results of overall sample.

| Variable | Model 1 | Model 2 | Model 3 | Model 4 | Model 5 |
|---|---|---|---|---|---|
| ICT | 0.0313 * (1.6839) | | 0.0324 * (1.7559) | 0.0305 (0.6319) | |
| TRANS | | −0.0901 ** (−2.4059) | −0.0916 ** (−2.4551) | | 0.2706 *** (3.8492) |
| ICT $^2$ | | | | 0.0011 (0.0195) | |
| TRANS $^2$ | | | | | −2.6839 *** (−5.9285) |
| INFL | 0.1614 * (1.8306) | 0.1308 (1.4869) | 0.1399 (1.5934) | 0.1614 * (1.8254) | 0.115 (1.3896) |
| UNEM | 0.4291 * (1.8895) | 0.3741 * (1.6563) | 0.3958 * (1.7567) | 0.4294 * (1.8824) | 0.2968 (1.3952) |

**Table 4.** *Cont.*

| Variable | Model 1 | Model 2 | Model 3 | Model 4 | Model 5 |
|---|---|---|---|---|---|
| GOV | −0.0273 | −0.033 | −0.033 | −0.0273 | −0.0038 |
| | (−0.7697) | (−0.9346) | (−0.9393) | (−0.7685) | (−0.1146) |
| $GOV_{t-1}$ | 0.1387 *** | 0.1492 *** | 0.1476 *** | 0.1387 *** | 0.1362 *** |
| | (4.4225) | (4.7529) | (4.7183) | (4.414) | (4.6019) |
| EDU | 0.0384 | 0.0201 | 0.0323 | 0.0384 | 0.0362 |
| | (0.8604) | (0.4587) | (0.7295) | (0.8589) | (0.8754) |
| $EDU_{t-1}$ | −0.0729 * | −0.0805 * | −0.0792 * | −0.0728 * | −0.0679 * |
| | (−1.6984) | (−1.8833) | (-1.8604) | (−1.6932) | (−1.6865) |
| FIV | 0.0039 | 0.0053 | 0.0051 | 0.0039 | −0.001 |
| | (0.3717) | (0.5102) | (0.4912) | (0.3715) | (−0.0997) |
| $FIV_{t-1}$ | −0.0138 | −0.0112 | −0.013 | −0.0138 | −0.0107 |
| | (−1.1333) | (−0.93) | (−1.0826) | (−1.1295) | (−0.9433) |
| $GINI_{t-1}$ | 0.4193 *** | 0.4521 *** | 0.4508 *** | 0.4193 *** | 0.5259 *** |
| | (8.1994) | (8.6197) | (8.6274) | (8.1671) | (10.34) |
| Constant | 0.0856 *** | 0.0928 *** | 0.0826 *** | 0.0857 *** | 0.0675 *** |
| | (5.024) | (5.8164) | (4.8842) | (4.667) | (4.3307) |
| Province | Yes | Yes | Yes | Yes | Yes |
| Year | Yes | Yes | Yes | Yes | Yes |
| Adjusted $R^2$ | 0.9766 | 0.9768 | 0.977 | 0.9765 | 0.9795 |
| Observations | 310 | 310 | 310 | 310 | 310 |
| Cross sections | 31 | 31 | 31 | 31 | 31 |
| *F* statistic | 263.7567 *** | 266.7793 *** | 263.6003 *** | 257.4878 *** | 296.4835 *** |
| *D.W.* statistic | 2.177 | 2.1462 | 2.1417 | 2.1771 | 2.2073 |

**Note:** This table reports results of panel regressions in which the dependent variable is Educational inequality (EduIneq). Independent variables are Information and communication technology (ICT) and Transportation infrastructure (TRANS). Controlled variables include Inflation (INFL), Unemployment rate (UNEM), Government intervention (GOV), Education expenditure (EDU), and Fixed assets investment (FIV). $ICT^2$ and $TRANS^2$ are the squares of ICT and Transportation infrastructure. Figures in parentheses are the t statistics of estimated coefficients; ***, **, * represent the significance at the levels of 1%, 5%, 10% respectively.

In the regression analysis of the overall sample, Model 1 and Model 2 explore the impact of ICT or transportation infrastructure on educational inequality. Model 3 explores the relationship among ICT, transportation infrastructure, and educational inequality based on Formula (2). Model 4 explores the nonlinear relationship between ICT and educational inequality based on Formula (3). Model 5 explores the nonlinear relationship between transportation infrastructure and educational inequality based on Formula (4).

By comparing the results of Model 1 and Model 2, it can be seen that there is a positive relationship between ICT and educational inequality, while transportation infrastructure has a negative impact on educational inequality. The results of Model 1 and Model 2 reveal that, under the condition of single factor, ICT could limit the availability of educational information, thus promoting the unequal distribution of educational resources, while transportation infrastructure would provide more educational opportunities for social groups while alleviating the rising inequality in the educational field. The results of Model 3 were similar to those of Model 1 and Model 2, indicating that ICT and transportation infrastructure can have significant effects on educational inequalities at the same time, but their effects are inverse, which supports Hypothesis 1a and Hypothesis 2a. In the results of Model 4, there was no significant nonlinear relationship between ICT and educational inequality. According to the results of Model 5, there is a significant inverted U-shaped relationship between transportation infrastructure and educational inequality, indicating that transportation infrastructure could promote educational inequality at some specific levels of transportation network development, and that the growth of transportation investment would strengthen the impact of transportation infrastructure on alleviating the unbalanced distribution of educational resources, which supports Hypothesis 2b.

### 4.2.2. Regression Analysis of the Overall Sample in Different Periods

The development of education depends on educational resources and educational policies, so education policy plays an important role in alleviating the problem of educational inequality. Given the changes of the education system in China, the government has enacted the reform policy of the education system since December 2010, and the goal of this policy is to adjust the distribution of educational resources, reducing the seriousness of educational inequality in different regions [48]. During the process of exploring the relationship among ICT, transportation infrastructure, and educational inequality, the overall sample was divided by the year of reform policy into different periods. The regression results of the overall sample in different periods are shown in Table 5.

**Table 5.** Regression results of overall sample in different periods.

| Variable | Model 6 | Model 7 | Model 8 | Model 9 | Model 10 | Model 11 |
|---|---|---|---|---|---|---|
| | 2006–2010 | | | 2011–2016 | | |
| ICT | 0.0051 (0.1147) | 0.1928 * (1.717) | | 0.0455 * (1.8011) | −0.0081 (−0.1226) | |
| TRANS | 0.0502 (0.6334) | | −0.2547 (−1.5926) | −0.1641 *** (−3.4059) | | 0.1093 (0.9853) |
| ICT $^2$ | | −0.2745 * (−1.7968) | | | 0.0788 (1.0013) | |
| TRANS $^2$ | | | 4.3522 ** (2.1778) | | | −1.7275 *** (−2.7973) |
| INFL | −0.0157 (−0.1417) | −0.0015 (−0.014) | 0.0471 (0.4229) | 0.2054 (1.19) | 0.3393 * (1.8913) | 0.1495 (0.8819) |
| UNEM | 0.4236 (0.6653) | 0.2909 (0.4651) | 0.4771 (0.778) | 0.1049 (0.3356) | 0.2719 (0.8367) | 0.1863 (0.602) |
| GOV | 0.045 (0.655) | 0.0465 (0.691) | 0.0615 (0.9161) | −0.0317 (−0.6104) | −0.0009 (−0.0174) | −0.0272 (−0.5319) |
| GOV$_{t-1}$ | 0.1029 * (1.7387) | 0.1145 ** (2.0093) | 0.0569 (0.9405) | 0.2128 *** (4.5077) | 0.2212 *** (4.5228) | 0.1921 *** (4.0673) |
| EDU | 0.0561 (0.3576) | −0.045 (−0.2971) | 0.1821 (1.1284) | 0.1153 ** (2.4156) | 0.1180 ** (2.3817) | 0.0953 ** (2.0663) |
| EDU$_{t-1}$ | 0.0045 (0.0274) | 0.0219 (0.1404) | −0.1307 (−0.7687) | −0.1038 ** (−2.2338) | −0.0946 * (−1.9657) | −0.0875 * (−1.889) |
| FIV | −0.0248 (−0.9096) | −0.0217 (−0.8071) | −0.0373 (−1.3752) | 0.0085 (0.6948) | 0.0065 (0.5043) | 0.0047 (0.3885) |
| FIV$_{t-1}$ | −0.0011 (−0.0422) | −0.0077 (−0.2893) | 0.0149 (0.5522) | −0.0003 (-0.0183) | −0.006 (−0.4041) | 0.0008 (0.0543) |
| GINI$_{t-1}$ | 0.3182 ** (2.2838) | 0.3382 ** (2.478) | 0.2438 * (1.7504) | 0.1919 ** (2.316) | 0.1051 (1.272) | 0.2829 *** (3.265) |
| Constant | 0.1156 ** (2.5366) | 0.099 ** (2.1602) | 0.1369 *** (3.132) | 0.1071 *** (4.1087) | 0.1117 *** (4.0045) | 0.1001 *** (3.9074) |
| Province | Yes | Yes | Yes | Yes | Yes | Yes |
| Year | Yes | Yes | Yes | Yes | Yes | Yes |
| Adjusted $R^2$ | 0.9791 | 0.9798 | 0.9803 | 0.9813 | 0.9799 | 0.9819 |
| Observations | 124 | 124 | 124 | 186 | 186 | 186 |
| Cross sections | 31 | 31 | 31 | 31 | 31 | 31 |
| *F* statistic | 131.7712 *** | 136.5271 *** | 139.7662 *** | 211.7463 *** | 196.633 *** | 218.6626 *** |
| *D.W.* statistic | 1.8393 | 1.8956 | 1.7956 | 2.251 | 2.2809 | 2.2521 |

Note: This table reports results of panel regressions in which the dependent variable is Educational inequality (EduIneq). Independent variables are Information and communication technology (ICT) and Transportation infrastructure (TRANS). Controlled variables include Inflation (INFL), Unemployment rate (UNEM), Government intervention (GOV), Education expenditure (EDU), and Fixed assets investment (FIV). ICT$^2$ and TRANS$^2$ are the square of ICT and Transportation infrastructure. Figures in parentheses are the t statistics of estimated coefficients; ***, **, * represent the significance at the level of 1%, 5%, 10% respectively.

According to the year of education system reform in China, we divided the overall sample into 2 sub-periods: pre-reform (2006–2010) and post-reform (2011–2016). Model 6, Model 7, and Model 8 were concentrated on the linear and nonlinear relationships between ICT, transportation infrastructure, and educational inequality from 2006 to 2010. Model 9, Model 10 and Model 11 were concentrated on these relationships from 2011 to 2016.

From the regression results of Model 6 and Model 9, it can be found that there was no significant relationship among ICT, transportation infrastructure, and educational inequality before the education system reform. However, after implementing the reform policy of education system, there was a significant positive correlation between ICT and educational inequality, while transportation infrastructure had a significant negative impact on educational inequality. The results of Model 6 and Model 9 demonstrate that the reform policy of education system has strengthened the relationship among ICT, transportation infrastructure, and educational inequality, and that educational policy could indirectly influence educational inequality from different aspects. By comparing the results of Model 7 and Model 10, there was a significant inverted U-shaped relationship between ICT and educational inequality before the education reform, while this relationship became insignificant after the implementation of the reform policy of education system, indicating that this reform policy has changed the impact mechanism of information technology on the distribution of educational resources. In comparison with the results of Model 8 and Model 11, there was no significant nonlinear relationship between transportation infrastructure and educational inequality during different periods.

4.2.3. Regression Analysis of the Overall Sample in Different Regions

In the application of educational resources, the geographical location and the economic development of different regions play an important role in improving the distribution of educational resources. Compared with other countries, China's financial resources and political resources are mostly concentrated in the eastern region, while the western region, as the economically backward area, gets fewer resources. Faced with this situation, the overall sample was divided into three subsamples based on the three major regions in China. The eastern regions included 11 provinces and cities, namely Beijing, Tianjin, Hebei, Liaoning, Shanghai, Jiangsu, Zhejiang, Fujian, Shandong, Guangdong, and Hainan. The central regions included 8 provinces and cities, namely Shanxi, Jilin, Heilongjiang, Anhui, Jiangxi, Henan, Hubei, and Hunan. The western regions included 12 provinces and cities in the western regions, including Inner Mongolia, Guangxi, Chongqing, Sichuan, Guizhou, Yunnan, Tibet, Shaanxi, Gansu, Qinghai, Ningxia, and Xinjiang. The regression results of the overall sample in different regions are shown in Table 6.

By comparing the results of Model 12, Model 15, and Model 18, there was no significant correlation between ICT and educational inequality in different subsamples, indicating that the development of information technology would not affect the unbalanced distribution of educational resources in three major regions. However, transportation infrastructure in the western regions has a significant negative impact on educational inequality, but the influence of transportation network is not significant in the eastern regions and the central regions. It is worth noting that the development of transportation infrastructure has brought more educational opportunities to economically backward regions, and improved the unbalanced distribution of educational resources, which supports the results of Chen et al. (2016) [39]. In addition, there was no significant non-linear relationship between ICT and educational inequality in three major regions, and the effects of ICT were similar to the results of overall sample. Through comparison with the results of Model 14, Model 17, and Model 20, there was a significant inverted U-shaped relationship between transportation infrastructure and educational inequality in the western regions, but this kind of relationship was not significant in other major regions, indicating that the development of transportation network could effectively improve the acquisition efficiency of external resources in economically backward areas, and significantly relieve the problem of educational inequality.

**Table 6.** Regression results of overall samples in different regions.

| Variable | Model 12 East | Model 13 East | Model 14 East | Model 15 Central | Model 16 Central | Model 17 Central | Model 18 West | Model 19 West | Model 20 West |
|---|---|---|---|---|---|---|---|---|---|
| ICT | −0.0252 (−1.0421) | −0.0856 (−1.1765) | | 0.0256 (0.6643) | 0.0012 (0.0083) | | 0.0891 (1.491) | 0.2854 (1.6586) | |
| TRANS | 0.0691 (0.8193) | | 0.0043 (0.018) | −0.0392 (−0.4836) | | −0.048 (−0.1963) | −0.145 ** (−2.493) | | 0.3635 *** (3.038) |
| ICT $^2$ | | 0.0639 (0.8791) | | | 0.0539 (0.1646) | | | −0.3954 (−1.1443) | |
| TRANS $^2$ | | | 1.159 (0.2888) | | | 0.1401 (0.0459) | | | −3.3157 *** (−4.7918) |
| INFL | −0.0951 (−0.6987) | −0.0887 (−0.6549) | −0.0823 (−0.6031) | 0.111 (0.5903) | 0.1285 (0.6789) | 0.1079 (0.5694) | 0.0945 (0.5849) | 0.1602 (0.9793) | 0.0804 (0.5517) |
| UNEM | 0.6449 * (1.9777) | 0.6692 ** (2.0655) | 0.7079 ** (2.1846) | 0.224 (0.832) | 0.2573 (0.9411) | 0.262 (0.986) | −0.9394 (−1.4902) | −0.835 (−1.2932) | −0.9465 * (−1.6854) |
| GOV | −0.0192 (−0.2339) | −0.0313 (−0.3906) | −0.0137 (−0.1654) | 0.1827 (1.6148) | 0.1774 (1.4927) | 0.1849 (1.6186) | −0.0935 * (−1.7418) | −0.0949 * (−1.7222) | −0.0564 (−1.1652) |
| GOV$_{t-1}$ | −0.0196 (−0.243) | −0.0054 (−0.0661) | −0.011 (−0.1357) | −0.1224 (−1.0888) | −0.1157 (−0.9768) | −0.1252 (−1.1099) | 0.1728 *** (3.6224) | 0.159 *** (3.2606) | 0.1433 *** (3.3604) |
| EDU | 0.0337 (0.4572) | 0.029 (0.3974) | 0.0483 (0.6607) | −0.1817 (−1.4996) | −0.1783 (−1.4323) | −0.1697 (−1.402) | 0.0671 (0.9034) | 0.0929 (1.2256) | 0.0645 (0.9699) |
| EDU$_{t-1}$ | −0.1075 (−1.5303) | −0.1206 (−1.6852) | −0.094 (−1.3436) | 0.1929 (1.4088) | 0.1849 (1.3215) | 0.2075 (1.5275) | −0.0436 (−0.6246) | −0.0196 (−0.2742) | −0.0039 (−0.0626) |
| FIV | 0.0161 (1.0692) | 0.0183 (1.2459) | 0.0172 (1.1386) | 0.0031 (0.1988) | 0.0034 (0.216) | 0.0018 (0.1114) | −0.0228 (−0.9623) | −0.029 (−1.1758) | −0.0198 (−0.9345) |
| FIV$_{t-1}$ | −0.0042 (−0.2163) | −0.0052 (−0.2702) | −0.0098 (−0.521) | −0.0098 (−0.5645) | −0.01 (−0.5735) | −0.0099 (−0.5608) | −0.0131 (−0.5311) | −0.0116 (−0.4532) | −0.0201 (−0.9053) |
| GINI$_{t-1}$ | 0.301 *** (2.7617) | 0.3164 *** (2.9107) | 0.3411 *** (3.3428) | 0.6433 *** (6.5566) | 0.633 *** (6.4889) | 0.6435 *** (6.4745) | 0.3372 *** (3.5907) | 0.278 *** (2.9921) | 0.4596 *** (5.1721) |
| Constant | 0.145 *** (3.933) | 0.1569 *** (4.0264) | 0.1197 *** (4.407) | 0.0499 * (1.725) | 0.0519 (1.6593) | 0.0527 * (1.7825) | 0.19 *** (4.3026) | 0.1735 *** (3.6318) | 0.1573 *** (3.9202) |
| Province | Yes | Yes | Yes | Yes | Yes | Yes | Yes | Yes | Yes |
| Yea | Yes | Yes | Yes | Yes | Yes | Yes | Yes | Yes | Yes |
| Adjusted $R^2$ | 0.9149 | 0.915 | 0.9138 | 0.9346 | 0.9343 | 0.934 | 0.9784 | 0.9772 | 0.9824 |
| Observations | 110 | 110 | 110 | 80 | 80 | 80 | 120 | 120 | 120 |
| Cross sections | 11 | 11 | 11 | 8 | 8 | 8 | 12 | 12 | 12 |
| *F* statistic | 40.0606 *** | 40.1151 *** | 39.5262 *** | 42.7898 *** | 42.6127 *** | 42.4153 *** | 174.7024 *** | 165.4577 *** | 215.5114 *** |
| *D.W.* statistic | 2.1522 | 2.1623 | 2.2006 | 2.4831 | 2.4182 | 2.5428 | 2.1181 | 2.2254 | 2.2564 |

Note: This table reports results of panel regressions in which the dependent variable is Educational inequality (EduIneq). Independent variables are Information and communication technology (ICT) and Transportation infrastructure (TRANS). Control variables include Inflation (INFL), Unemployment rate (UNEM), Government intervention (GOV), Education expenditure (EDU), and Fixed assets investment (FIV). ICT$^2$ and TRANS$^2$ are the square of ICT and Transportation infrastructure. Figures in parentheses are the t statistics of estimated coefficients; ***, **, * represent the significance at the level of 1%, 5%, 10% respectively.

## 5. Discussion

From the results of the empirical analysis, the relationships among ICT, transportation infrastructure, and educational inequality were different in the overall sample and the subsamples. In previous studies on ICT, the affordability of information technology has been shown to change the role of ICT in the educational field [18,24]. However, transportation infrastructure as an important factor in promoting the economic development can bring more opportunities for the labor force [39,46,47]. Based on the results of the overall samples, the findings provide insight into the impact of ICT on educational inequality, which may be affected by the affordability of information services. The allocation of educational resources will be more unreasonable, and the results of ICT's impact were similar to the findings of Ayanso and Lertwachara (2015) [24]. ICT changes the utilization efficiency of educational resources, and helps some learners to obtain more educational information. However, the affordability of information technology can widen the gap between users in the process of resource acquisition, and this situation may also promote inequality of educational outcomes, just like the discussions of ICT in Angus et al. (2004) [18]. It is worth noting that the development of transportation infrastructure can provide more educational opportunities for the labor force, thus alleviating educational inequality, and this is similar to the results of job accessibility in Schaffer and Siegele (2009) [30]. Compared to the impact of information technology, transportation infrastructure could shorten the distance between regions, and provide better transportation services for labor migration [39]. Unlike previous research on transportation infrastructure, the nonlinear relationship between transportation infrastructure and educational inequality was tested in this paper. Combined with the inverted U-shaped relationship, it was found that the sustained growth of transportation investment would bring more educational opportunities to social groups, and reduce the adverse impact of educational inequality from a long term perspective.

While exploring the impact of educational reform policy, the roles of ICT and transportation infrastructure in educational field are changed. The goal of education reform is to solve the problem of educational inequality, and educational policy can influence the distribution of educational resources from different areas [4]. Since the Chinese education system reform, government regulators have put more effort into the allocation of educational resources [2]. From the results of overall samples in different periods, the efforts of regulators raise the limitations of information services for the availability of educational resources, which supports the findings of Braga et al. (2013) [4]. It is interesting to note that the impact mechanism of ICT has changed from the nonlinear type into the linear type, and this finding can be explained by the results of the digital divide in Chao and Yu (2016) [25]. Moreover, the conflict between the hukou system and educational equality is an important problem, while the development of the transportation network can alleviate the restrictions of the hukou system on educational opportunities, which is similar to the results of Golley and Kong (2018) [2]. In the results of China's three major regions, the role of transportation infrastructure in economically backward regions was very important, and should not be ignored by government regulators and researchers. Compared with other regions, economic development in the western regions has been restricted by external resources for a long time, and most human resources in the western regions have moved to other major regions, so it is difficult to develop and upgrade the real industry [46]. Faced with this difficulty, the growth of transportation investment could significantly improve the acquisition efficiency of external resources in the western regions, and the industrial structure would be upgraded accordingly. From the perspective of educational opportunities, the hukou system has a stronger limitation on educational groups in the eastern and central regions [2]. However, the development of the transportation network would bring more educational opportunities to the western regions, and improve the utilization efficiency of educational resources. Compared with the results of Yu and De (2012), the findings of transportation infrastructure in different regions can help government regulators to design the population policy, and transportation investment will motivate the allocation of essential educational resources [47].

## 6. Conclusions and Recommendations

Educational attainment is the basic element of human capital, while the unbalanced distribution of educational resources can lead to the problem of educational inequality. At the same time, educational inequality has hindered the development of education, and this kind of inequality directly influences the development of regional economy. In developing countries, government regulators have tried to upgrade the industrial structure from labor-intensive to technology-intensive, but this sustainable development strategy has been limited by educational inequality. Considering the difficulties of industrial transformation, many countries still depend on labor-intensive industries, while human capacity struggles to match the development of production technology. The labor market is an important factor in improving the output of economic activities, but educational inequality may limit the skills of the labor force.

Alleviating the problem of educational inequality requires considerable time, and needs to consider many factors from different dimensions. Based on a sample of 31 provinces in China from 2006 to 2016, we constructed panel data models to analyze the linear and nonlinear relationships among ICT, transportation infrastructure, and educational inequality. Combined with the characteristics of economic development in China, ICT could bring convenient information services to users, but such services limit the acquisition efficiency of information resources for some users. The affordability of information technology widens the gap between users in terms of information acquisition, which reduces the generality and availability of educational resources. On the other hand, transportation infrastructure could effectively shorten the distance between cities and reduce geographical restrictions on educational opportunities. Considering China's population policy, the hukou system has restrained the labor force to obtain equal educational opportunities for a long time, while the development of transportation network can largely alleviate this problem. In addition, the effect of transportation infrastructure on educational inequality would increase with the growth of transportation investment, reflecting the important role of the transportation network in resource allocation.

After the implementation of the education system reform in China, the roles of ICT and transportation infrastructure in educational field have been strengthened, indicating that the reform policy could enhance the relationship among information technology, transportation network, and educational resources. In addition, the education system reform changes the impact mechanism of ICT on educational inequality, which helps government regulators flexibly control the effect of ICT. Furthermore, transportation infrastructure in the western regions could significantly alleviate the seriousness of educational inequality, and its effect would increase the growth of the transportation investment. Considering the characteristics of different major regions, the western regions, as the economically backward areas, have long suffered from the problem of labor loss, while the development of transportation network brings more external resources to the western regions, guaranteeing the regional economic development.

There are some limitations of our work in terms of the research design and the empirical analysis. For the research design, the growth of transportation investment was used to measure the variable of transportation infrastructure, but measurement methods of transportation density and transportation capital were not selected in this paper. Considering the economic environment in China, transportation investment could more directly represent the development status of transportation network. For the empirical analysis, different subsamples were obtained according to the China's three major regions, but the results could not demonstrate the relationship between ICT and educational inequality in the eastern regions and central regions. Future research could take into account the different measurements of transportation infrastructure, and this method of variable construction will match the characteristics of economic development for various countries. Moreover, the impact of educational reform policy on ICT is important for studies on educational inequality, and this should be the starting point for the future related studies. Furthermore, different division methods of subsamples can be used to explore the internal characteristics of overall sample, which will provide a comprehensive explanation for the emergence of educational inequality.

**Author Contributions:** Writing—original draft preparation, P.Z. and F.C.; Writing—review and editing, W.W. and P.S.; Formal analysis, W.W.; Data curation, C.Z.

**Funding:** This research was funded by the Key Project of National Social Science Fund "Informatization Promotes the Equity of Basic Education in The New Era", grant number 18ZDA334; the Project of Science and Technology Development Institute, grant number CCNU18KFY04; the China Scholarship Council, grant number 201806775045; the Project of Fundamental Research Funds for the Central Universities, grant number 2019CDSKXYJG0037; the Project of Research on the Cultivation Model of Entrepreneurship Ability of College Students in Western China, grant number 2017-GX-002; the Project of Research on Innovative Thinking Design and Cultivation Model of Entrepreneurship Ability of Universities in Chongqing, grant number 183128.

**Conflicts of Interest:** The authors declare no conflict of interest.

## Appendix A

**Table A1.** Results of the unit root test.

| Variables | LLC | IPS | ADF-Fisher | PP-Fisher | Stable or Not |
|---|---|---|---|---|---|
| EduIneq | −6.1363 *** | 1.3983 | 46.6849 | 94.6206 *** | No |
| ICT | −8.3802 *** | −1.4766 | 82.569 ** | 112.274 *** | No |
| TRANS | −10.8197 *** | −3.5485 *** | 125.39 *** | 138.475 *** | Yes |
| INFL | −35.1408 *** | −9.0169 *** | 244.058 *** | 449.23 *** | Yes |
| UNEM | −9.5518 *** | −1.7777 ** | 98.8156 *** | 101.764 *** | Yes |
| GOV | −10.0648 *** | −1.1293 | 78.9216 | 53.2447 | No |
| EDU | −5.4649 *** | −0.4543 | 68.6602 | 51.8042 | No |
| FIV | −10.5637 *** | −1.3802 | 88.161 ** | 94.6012 *** | Yes |
| ΔEduIneq | −18.6069 *** | −4.6505 *** | 179.14 *** | 304.595 *** | Yes |
| ΔICT | −14.3184 *** | −3.8374 *** | 156.852 *** | 276.858 *** | Yes |
| ΔTRANS | −19.3405 *** | −5.1003 *** | 188.87 *** | 295.705 *** | Yes |
| ΔINFL | −46.0877 *** | −9.1431 *** | 296.054 *** | 385.31 *** | Yes |
| ΔUNEM | −17.6622 *** | −2.4659 *** | 120.753 *** | 125.128 *** | Yes |
| ΔGOV | −14.9639 *** | −1.9267 ** | 108.604 *** | 153.813 *** | Yes |
| ΔEDU | −13.22 *** | −2.2245 ** | 119.807 *** | 151.604 *** | Yes |
| ΔFIV | −17.2492 *** | −4.4288 *** | 175.484 *** | 293.804 *** | Yes |

Note: ***, ** represents the significance at the level of 1%, 5% respectively; Δ indicates the first-order difference processing on the variable.

**Table A2.** Results of discriminating the panel data model type.

| Test Method | Hypothesis | Statistic | *p* Value |
|---|---|---|---|
| F test | H0: The true model is a mixed regression model<br>H1: The true model is a fixed effects model | 3.0362 | 0.0000 |
| Hausman test | H0: The true model is a random effect model<br>H1: The true model is a fixed effects model | 86.9791 | 0.0000 |

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
