# Peer review of "Does the Development of Information and Communication Technology and Transportation Infrastructure Affect China’s Educational Inequality?"

_sustainability, doi:10.3390/su11092535_

Round 1
Reviewer 1 Report
LINE | COMMENT |
Title | Spell out ICT for those that do not know what it is |
18 | “the whole society” sounds awkward |
19 | No “the” before sustainable |
Entire manuscript | This manuscript would benefit from having a native English speaker review the grammar, especially the word choice. It feels like it was put through a translator and is quite awkward at times |
24 | ICT promotes having unequal distribution? |
Abstract | The abstract is very detailed and, at times, difficult to read. May want to make the abstract more accessible and readable to those not familiar with this area of research to help interest new readers |
Intro | Be sure to define your key elements, including human capital, educational attainment, and economic output. It helps those outside the field and helps to make sure we are all on the same definition |
Introduction | The intro is used to set the stage, so you don’t really need to say was this paper will be doing or what the results can do. It isn’t an overview of the paper, but more like a way to give an overview of the field, followed by specifics in section 2 |
130 | Title can just be “Educational Inequality”, we know it is the background |
Entire manuscript | It would help to look for run on sentences. Some are 4 lines long and try to include many ideas. The manuscript also feels like it is repeating itself over and over again, so please look it over to be clear and concise. |
Lit review | You do not need to include your null hypotheses in a paper |
Entire manuscript | Be sure to write everything in the same tense, not a mix of present and past |
3.1 | Please explain what types of data are in these yearbooks as not all countries use them |
3.2.4 | Each paragraph should be a subsection as it is written, so please format it as such |
Methods | Please be more clear with where the data was originally collected into the yearbooks, etc, to help the reader |
Results | Tables are way too big and difficult to read. Please choose the most important information – readers do not need everything. You as the writer need to decide what info we need to understand your claims |
Discussion? | After results there should be a separate discussion about what the results mean to you, the field, etc. This is where you support your claim, not in the results or conclusion. If combining with Results, add the name to the section title. The conclusion is an overview of what was found, what it means for future research, etc., not the place to make claims |
Conclusion | Be careful about writing in the absolute |
Overall an interesting paper, but there was just so much information and grammatical issues that it really detracted from the research. It would really help to make the paper more clear and concise. Either explain your field jargon or remove it, and check the grammar to ensure it is saying what you actually want it to say.
Author Response
Point 1: In title, spell out ICT for those that do not know what it is.
Response 1: Thanks for reviewer’s suggestion on the title of our manuscript. The suggestion is very important for our work, and it is not the normal way of using Information and Communication Technology’s abbreviation (labelled as ICT) in the previous title. In former manuscript, using ICT in the title brings many difficulties for readers to understand, and this cannot show the core content of our work. Spelling out ICT can show the research content directly and clearly.
At last, thanks again for the suggestion provided by the reviewer. We use “information and communication technology” to replace ICT in the title of revised manuscript. Please refer to the revised title.
Point 2: In line 18, “the whole society” sounds awkward.
Response 2: Thanks to reviewer for the valuable suggestion of word usage. We hope to express the relationship among ICT, transportation infrastructure and educational inequality from the macro perspective, and solving the influence of educational inequality is the starting point of our work. For the standard usage, we use “the public” to replace “the whole society” in the revised manuscript, and then help readers understand the perspective of our research. Please refer to the revised abstract and section 1, 6 for the detailed modifications.
Point 3: In line 19, no “the” before sustainable.
Response 3: Thanks for reviewer’s suggestion on the definite article. In former manuscript, we don’t consider the definite article of sustainable carefully. According to the reviewer’s suggestion, the collocation of sustainable is quite special, so we revise this problem in many places of revised manuscript. Please refer to the revised abstract and section 1 for the detailed modifications.
Point 4: This manuscript would benefit from having a native English speaker review the grammar, especially the word choice. It feels like it was put through a translator and is quite awkward at times.
Response 4: Thanks to reviewer for the valuable suggestion on the English expression of our work. This paper takes a lot of time for our team, and the research topic is discussed and revised many times. We hope to alleviate the problem of educational inequality in developing countries, especially in China. In the process of writing, there are some problems for the word choice and the sentence expression, and they also bring many difficulties for readers to understand. For revising the manuscript, a native English speaker helps us to review the full text of our work. We do a lot of modifications for the word choice and the sentence expression.
At last, thanks again for this important suggestion provided by the reviewer. Please refer to the revised manuscript for the detailed modifications.
Point 5: In line 24, ICT promotes having unequal distribution?
Response 5: Thanks to reviewer for questioning this finding of our work. In the literature review, the advantage of ICT is to provide some learners with convenient services, and this may also affect the educational quality. In order to explore the relationship between ICT and educational inequality, the data of China is used for empirical analysis, and there is a significant positive relationship between ICT and educational inequality. In former abstract, we propose that ICT promotes unequal distribution of educational resources, but this cannot display our actual findings. In order to better demonstrate the value of findings, we propose that there is a positive relationship between ICT and educational inequality in the revised abstract, revealing that ICT may lead to this problem of education. Please refer to the revised abstract for the detailed modifications
Point 6: The abstract is very detailed and, at times, difficult to read. May want to make the abstract more accessible and readable to those not familiar with this area of research to help interest new readers.
Response 6: Thanks for reviewer’s suggestion on the abstract. This suggestion is very important for our paper. In former manuscript, we add a lot of content in the abstract, which makes this part very difficult to read. In order to better show findings of our research, we revise the part of the abstract and delete the redundant content. In addition, we modify the presentation of the abstract by using the shorter sentences to show the value of our findings.
All in all, it is grateful to receive reviewer’s suggestion on the abstract, which is the core of our work. Please refer to the revised abstract for the detailed modifications
Point 7: In introduction, be sure to define your key elements, including human capital, educational attainment, and economic output. It helps those outside the field and helps to make sure we are all on the same definition.
Response 7: Thanks to reviewer for the comments on some key elements in introduction. During introducing the research background, we neglect the explanation of some key elements, and this makes the content of the introduction difficult to state the importance of our research. In the revised manuscript, we explain these key elements respectively, including human capital, educational attainment and economic output. In addition, we use simple and clear sentences to explain these elements, and this expression can provide better understanding of our work for readers in different fields. Please refer to the revised section 1 for the detailed modifications.
Point 8: The intro is used to set the stage, so you don’t really need to say was this paper will be doing or what the results can do. It isn’t an overview of the paper, but more like a way to give an overview of the field, followed by specifics in section 2.
Response 8: Thanks for reviewer’s suggestion on the introduction. This suggestion provides a clear idea for our revision, and also brings the standard structure of the introduction. We delete the research models and the research results in the introduction, and simplify the content of this section. In addition, we retain the contributions of our work in the last part of introduction, so readers can understand the importance of our research easily. Please refer to the revised section 1 for the detailed modifications.
Point 9: In line 130, title can just be “Educational Inequality”, we know it is the background.
Response 9: Thanks to reviewer for the suggestion on the literature review. This suggestion can help us to obtain a better structure of the literature review. In former manuscripts, we fear that the title of section 2.1 would be same as that of section 3.2.1, which leads to unnecessary misunderstanding for readers. Based on reviewer’s suggestion, we change the title of section 2.1 to “Educational inequality”, and set the title of section 3.2.1 to “Dependent variable”. The new structure of literature review can better show the content of this section. Please refer to the revised section 2.1 for the detailed modifications.
Point 10: It would help to look for run on sentences. Some are 4 lines long and try to include many ideas. The manuscript also feels like it is repeating itself over and over again, so please look it over to be clear and concise.
Response 10: Thanks for reviewer’s suggestion on the expression of our work. After checking former manuscript, we find that there are some repetition problems of statements, and some statements are too long. These problems are mainly concentrated in the section of introduction and conclusion. We revise these problems of the entire manuscript accordingly, and simplify many long statements with deleting the repetitive ones. Please refer to the revised section 1 and 6 for the detailed modifications.
Point 11: In literature review, you do not need to include your null hypotheses in a paper
Response 11: Thanks to reviewer for the suggestion on the hypotheses in literature review. In this part, we purpose the research hypotheses, and hope that these hypotheses can be tested in the empirical process. We retain the null hypothesis in this section, because we find that there are some interesting results in different periods subsamples for testing this hypothesis, which is also an important direction for our future research.
Point 12: Be sure to write everything in the same tense, not a mix of present and past.
Response 12: Thanks for reviewer’s suggestion on the tense of our work. This suggestion improves our paper greatly, and we don’t notice this problem in former manuscript. After checking the entire manuscript, we modify the tense of our work and keep all the sentences in the same tense. In addition, we use the past tense in literature review to better distinguish the existing research results from our research. Please refer to the revised section 2, 3 and 6 for the detailed modifications.
Point 13: In section3.1, please explain what types of data are in these yearbooks as not all countries use them.
Response 13: Thanks for reviewer’s suggestion on the data source. In this paper, the data mainly come from China Statistical Bureau and various kinds of statistical yearbooks, including education statistics yearbook, transportation statistics yearbook and financial yearbook. In order to provide a better understanding of the data, we add some introductions of these yearbooks in section 3.1, and explain the types of these data. In addition, we introduce the data used by the core variables, in order to show the characteristics of the data. Please refer to the revised section 3.1 for the detailed modifications.
Point 14: In section 3.2.4, each paragraph should be a subsection as it is written, so please format it as such.
Response 14: Thanks for reviewer’s suggestion on the subsection of control variables. In former manuscript, section 3.2.4 introduces all control variables. We modify the structure of variable construction and explain the control variables in section 3.2.3. Based on the reviewer’s suggestion, we add a subsection to each control variable. Please refer to the revised section 3.2.3 for the detailed modifications.
Point 15: In methods, please be more clear with where the data was originally collected into the yearbooks, etc, to help the reader.
Response 15: Thanks for reviewer’s suggestion on the data in methods. With the modifications in section 3.1, we explain the data used by the variables and introduce the relationship between the data and the yearbooks. In the variable construction of Section 3.2, the data used for each variable come from China Statistical Bureau and various yearbooks in Section 3.1. Please refer to the revised section 3.1 and 3.2 for the detailed modifications.
Point 16: In results, tables are way too big and difficult to read. Please choose the most important information – readers do not need everything. You as the writer need to decide what info we need to understand your claims.
Response 16: Thanks to reviewer for the suggestion on the tables in results. In research results, there are many interesting findings, which are mainly obtained by comparing different variables or different subsamples. Table 6 is a very big table, but the results in this table can demonstrate the regional differences of China better. We hope that more results can be retained in the tables, which can support the results of our work. In addition, we add variable explanations into each table, and this can help readers understand the tables of results better. Please refer to the revised section 4.2.1, 4.2.2 and 4.2.3 for the detailed modifications.
Point 17: After results there should be a separate discussion about what the results mean to you, the field, etc. This is where you support your claim, not in the results or conclusion. If combining with Results, add the name to the section title. The conclusion is an overview of what was found, what it means for future research, etc., not the place to make claims.
Response 17: Thanks to reviewer for the suggestion on the separate discussion. This suggestion is crucial for our paper. We add section 5 before the conclusions, and set the discussion of empirical results in this section. The separate discussion can better show the important findings of our work, which also makes the content of analysis and discussion clearer. We change section 5 to introduce the discussion of results, and explain the findings from the empirical analysis. The new structure of manuscript will be more standardized and complete. Please refer to the revised section 5 for the detailed modifications.
Point 18: In conclusion, be careful about writing in the absolute.
Response 18: Thanks for reviewer’s suggestion on the conclusion. We make a lot of modifications to the content of the conclusion and deleted a lot of redundant sentences. In this section, we focus on the findings of our work, which also provide corresponding suggestions for government regulators and researchers. In addition, we revise the expression of some sentences in conclusion to avoid the problem of writing in the absolute. Please refer to the revised section 6 for the detailed modifications.

Reviewer 2 Report
There was a great deal of repetition in the manuscript. It seems everything is affected by everything else, i.e., transportation infrastructure, educational inequality; and distribution of educational resources. ICT could widen the gap between the urban and rural areas. People leaving the rural areas would make a difference in the whole society.
“Information technology plays an important role in the field of education, but the relationship between ICT and educational inequality is still at the exploratory stage.”
The tables are carefully and clearly explained. The variables are explained well.
Authors state government regulators need to pay more attention to the role of ICT in the educational field. Secondly the role of transportation infrastructure should not be ignored because of the impact of population policies in China.
There were missing words like “a” or “the” ect that made me stop and re-read the section I was reading. Not a big deal, just a little irritating.
Author Response
Point 1: There was a great deal of repetition in the manuscript. It seems everything is affected by everything else, i.e., transportation infrastructure, educational inequality; and distribution of educational resources. ICT could widen the gap between the urban and rural areas. People leaving the rural areas would make a difference in the whole society.
Response 1: Thanks to reviewer for questioning the relationship among ICT, transportation infrastructure and educational inequality. In our work, the core of research content is to explore the influence factors of educational inequality, and we choose the information dimension and the transportation dimension as the starting point. Not all learners can take the advantages of ICT, and this may enlarge the inequality between users in the resource acquisition. In this manuscript, we discuss the impact of ICT on educational inequality, and there are some interesting findings that ICT may aggravate education inequality, but the influence mechanism of ICT can be changed by the educational policy.
Point 2: “Information technology plays an important role in the field of education, but the relationship between ICT and educational inequality is still at the exploratory stage.”
Response 2: Thanks for reviewer’s question on literature review. In some existing researches, ICT has been an efficient educational tool for a long time, and provides many platforms for learners. However, from the affordability of information technology, ICT cannot meet the needs of all learners, which also makes the educational quality difficult to be guaranteed. Some scholars have proposed the concept of digital inequality, and this will be directly reflected in the field of education.
Point 3: There were missing words like “a” or “the” ect that made me stop and re-read the section I was reading. Not a big deal, just a little irritating.
Response 3: Thanks for reviewer’s suggestion on the definite article. After checking the entire manuscript, we revise the sentences with missing words and adjust the usage of “a” and “the”. In the process of revision, we find that the lack of definite articles will affect the understanding of readers, and these problems will make our paper difficult to read. Please to refer the revised section 1, 2, 5 and 6 for the detailed modifications.

Reviewer 3 Report
I recommend adding bibliographic indices to lines 131-141 because you mentioned previous research.
Lines 277-283: repeat idea, recommend deletion.
I recommend rewriting the title to the table 2. It is not clear.
Under Table 1, 2,3,4,5. I recommend mentioning the meaning of the acronyms in the tables.
Table 2,3,4,5. I recommend rewriting their title, more specific.
Line 590-593 I recommend deleting, repeating idea, is not a conclusion.
I recommend that the conclusions section be reviewed, specific and focused on the topic of research and the results obtained.
Author Response
Point 1: I recommend adding bibliographic indices to lines 131-141 because you mentioned previous research.
Response 1: Thanks to reviewer’s suggestion on the bibliographic indices. In the literature review, we neglect the citation at the beginning part, and this makes the literature review to be lack of standardization. We add the corresponding bibliographic indices to lines 131-141, and adjust the order of references. Please refer to the revised section 2 and references for the detailed modifications.
Point 2: Lines 277-283: repeat idea, recommend deletion.
Response 2: Thanks for reviewer’s suggestion on the repeat idea. After checking these lines, we delete these sentences, and change the expression of other sentences like this problem. In former manuscript, there are some repetitive sentences, and we simplify these sentences for improving the quality of writing. Please refer to the revised section 1, 5 and 6 for the detailed modifications.
Point 3: I recommend rewriting the title to the table 2. It is not clear.
Response 3: Thanks for reviewer’s suggestion on the title of table 2. We revise the title of table 2, and replace the original title with “Variable definitions and measures”. The new title of this table can show the contents of variables, which also helps readers understand what we do. Please refer to the revised section 3.2.3 for the detailed modifications.
Point 4: Under Table 1, 2,3,4,5. I recommend mentioning the meaning of the acronyms in the tables.
Response 4: Thanks to reviewer’s suggestion on the explanation of the acronyms in the tables. We add the meaning of the acronyms to these tables, and explain the types of variables in detail. The revised tables can display the analysis content more directly, and help readers better understand the research results in different tables. Adding the meaning of the acronyms into the tables can support our empirical results. Please refer to the revised section 3 and 4 for the detailed modifications.
Point 5: Table 2,3,4,5. I recommend rewriting their title, more specific.
Response 5: Thanks for reviewer’s suggestion on the title of these tables. We revise the titles of table 2 and table 3. The new titles can show the contents of the tables more specific. Considering the information of table 4 and table 5, we hope to retain the titles of these tables, and the titles of table 4 and table 5 will show the empirical results obtained in different analysis processes. Please refer to the revised section 3 and 4 for the detailed modifications.
Point 6: Line 590-593 I recommend deleting, repeating idea, is not a conclusion.
Response 6: Thanks for reviewer’s suggestion on the repeat idea. We delete the repetitive ideas in line 590-593. In addition, we adjust the content of the conclusion, and change the expression of many sentences. In the structure of the conclusion, we delete the relevant content of the empirical process and simplify some research findings. Please refer to the revised section 6 for the detailed modifications.
Point 7: I recommend that the conclusions section be reviewed, specific and focused on the topic of research and the results obtained.
Response 7: Thanks to reviewer’s suggestion on the conclusion. This suggestion is very important to our work. In former manuscript, the conclusions section contains a lot of redundant contents, which cannot intuitively reveal the value of the research results. Based on the reviewer’s suggestion, we simplify the contents of the conclusions, and make the conclusion to be more focused on the research topic. In addition, we add the discussion of empirical results in section 5 for supporting the conclusions section.
All in all, it is grateful to receive reviewer’s suggestion on the core of the conclusion, which could be beneficial to our future research. Please refer to the revised section 6 for the detailed modifications.
